# The critical role of infection prevention overlooked in Ethiopia, only one-half of health-care workers had safe practice: A systematic review and meta-analysis

**Biniyam Sahiledengle** *, **Yohannes Tekalegn**, **Demelash Woldeyohannes**

Department of Public Health, MaddaWalabu University Goba Referral Hospital, Bale Goba, Ethiopia

* biniyam.sahiledengle@gmail.com

## Abstract

### Background

Effective infection prevention and control measures, such as proper hand hygiene, the use of personal protective equipment, instrument processing, and safe injection practicein the healthcare facilitiesare essential elements of patient safety and lead to optimal patient out-comes. In Ethiopia, findings regarding infection prevention practices among healthcare workers have been highly variable and uncertain. This systematic review and meta-analysis estimates the pooled prevalence of safe infection prevention practices and summarizesthe associated factors among healthcare workers in Ethiopia.

### Methods

PubMed, Science Direct, Google Scholar, and the Cochran library were systematically searched. We included all observational studies reporting the prevalence of safe infection prevention practices among healthcare workers in Ethiopia. Two authors independently extracted all necessary data using a standardized data extraction format. Qualitative and quantitative analyseswere employed. The Cochran Q test statistics and $I^2$ tests were used to assess the heterogeneity of the studies. A random-effects meta-analysis model was used to estimate the pooled prevalence of safe infection prevention practice.

### Results

Of the 187 articles identified through our search, 10 studies fulfilled the inclusion criteria and were included in the meta-analysis. The pooled prevalence of safe infection prevention practice in Ethiopia was 52.2% (95%CI: 40.9–63.4). The highest prevalence of safe practice was observed in Addis Ababa (capital city) 66.2% (95%CI: 60.6–71.8), followed by Amhara region 54.6% (95%CI: 51.1–58.1), and then Oromia region 48.5% (95%CI: 24.2–72.8), and the least safe practices were reported from South Nation Nationalities and People (SNNP) and Tigray regions with a pooled prevalence of 39.4% (95%CI: 13.9–64.8). In our qualitative syntheses, the odds of safe infection prevention practice were higher among healthcare

**Data Availability Statement:** All relevant data are within the manuscript and its Supporting information files.

**Funding:** The author(s) received no specific funding for this work.

**Competing interests:** The authors have declared that no competing interests exist.

workers who had good knowledge and a positive attitude towards infection prevention. Also, healthcare workers working in facilities with continuous running water supply, having infection prevention guideline, and those received training were significantly associated with-higher odds of safe infection prevention practice.

## Conclusions

Infection prevention practices in Ethiopia was poor, with only half of the healthcare workers reporting safe practices. Further, the study found out that there were regional and professional variations in the prevalence of safe infection prevention practices. Therefore, the need to step-up efforts to intensify the current national infection prevention and patient safety initiative as key policy direction is strongly recommended, along with more attempts to increase healthcare worker's adherence towards infection prevention guidelines.

## Background

Infection prevention and control is a set of practices, protocols, and procedures that are put in place to prevent infections that are associated with the healthcare system. Effective infection prevention and control measures, such as proper hand hygiene, the use of personal protective equipment (PPE), environmental cleaning, instrument processing, safe injection, and safe disposal of infectious wastes in the healthcare facilitiesmaximize patient outcomes and are essential to providing effective, efficient, and quality health care services [1–3]. Healthcare workers (HCWs) compliance with these recommended measures is termed as safe infection prevention practice.

Worldwide, healthcare-acquired infections (HAIs) affecting the quality of care of hundreds of millions of patients every year, contributing to increased morbidity, mortality, and substantial healthcare cost [1,2,4,5]. According to the World Health Organization (WHO), at any point in time forevery hundred hospitalized patients, ten will acquire at least one HAI [3]. The Centre for Disease Prevention and Control (CDC) estimates 2 million patients who will suffer from HAIs every year in the United States (US), and nearly one hundred thousand of them die [5], costing as much as 4.25 billion United States dollars [6]. Studies conducted in low-income settings showed that the prevalence of HAIs varies from 5.7% to 19.1%, with a pooled prevalence of 10.1% [7]; and the cumulative incidence range from 5.7% to 45.8% [8]. Further,in many cases,adherence towards infection prevention recommendations among healthcare workers (HCWs) in many low-income settings in general is poor [9–13].

In Ethiopia, the burden of HAIs is a major public health problem with a significant impact on hospitalized patients [14–16]. According to the finding of some pocket studies a high prevalence of HAIs has been reported from all corners of the country from 15.4% in north Ethiopia [15], 11.4%-19.4% in southwest Ethiopia [16,17], to 16.4% in central Ethiopia [18]. Although, a large proportion of HAIs can be prevented with inexpensive and cost effectiveinfection prevention and control measures;the evidence available suggests that healthcare facilities in Ethiopia do not have effective infection control programs [9]. In addition,HCWs compliance towards infection prevention and control (IPC) measures are critically low and a potential common problem in the country [9,19,20].

There is evidence that demonstrates the role of HCWs infection preventioncompliance on the reduction of HAIs [21–23] for example, Sickbert et al, in their study reported an

improvement in hand hygiene compliance of healthcare workers by 10%, which associated with a significant reduction in overall HAIs [22]. According to the World Health Organization (WHO) report, it is estimated that effective infection prevention and control (IPC) measures reduce HAIs by at least 30% [21]. In this context, adherence to the recommended infection prevention and patient safety practice is the best way in preventing patients, healthcare workers, and communities at large from HAIs. And the long-term solution to reduce the problems of HAIslie on actions to implement effective IPC measures in healthcare facilities [3,9,10,13]. Despite these facts-in many low-income settings, with healthcare systems and resources similar to Ethiopia,lack of well-trained HCWs, lack of infection prevention and control policies, and lack of technical guidelines consistent with the available evidence essential to provide a robust framework to support the performance of good IPC practices made the promotion of IPC practices a bit challenging [9,15,24–28].

To maximize the prevention of HAIs in Ethiopia, there has been a growing recognition of the need for safe infection prevention practice at all levels. Since the publication of the second Ethiopia National Infection Prevention Guidelines in 2012 [9], considerable progress has been made in understanding the basic principles, acceptance and use of evidence based Infection Prevention (IP) practices, including Clean and Safe Hospital (CASH), Clean Care is Safer Care campaigns, and Initiatives-Saving Lives through Safe Surgery (SaLTS). The national Infection Prevention and Patient Safety (IPPS) manual serve as a standardized IP reference manual for healthcare providersin all healthcare delivery systems. Also, it is intended to serve HCWs by providing clear guidance in the provisions of standard infection prevention and patient safety practices. The key components in the manual include standard precautions, hand hygiene, personal protective equipment, safe injection practice, processing instrument, and healthcare waste management [9]. Importantly, the existence of the IPC guidelines alone is not sufficient to ensure compliance and implementation of IPC recommendations; and findings clearly indicate that HCWs compliance is a prerequisite for successful guideline adoption. Previously conducted primary studies reported inconsistent findings regarding HCWs infection prevention practice in Ethiopia [19,20,27,29–33]. For instance, a study done in southeast Ethiopia showed that only 36.3% of HCWs had safe infection prevention practice [20], 15.0% in southern Ethiopia [33], 66.1% in central Ethiopia [27], andin northern Ethiopia42.9% of HCWs had acceptable practice [19]. Although the reporting of such practices is important for the prevention and control of HAIs and improving quality of care, the existed studies had many differences in the geographical regions and preceded remarkable variations in the reported practices. Due to the aforementioned reason, we conducted a systematic review and meta-analysis of observational studies to estimate the pooled prevalence of safe infection prevention practices among HCWs in Ethiopia. Also, we aim to summarize descriptively the factors that were associated with safe infection prevention practice.

## Materials and methods

### Search strategy

The protocol for this review was registered in the International Prospective Register of Systematic Reviews (PROSPERO), the University of York Centre for Reviews and Dissemination (record ID: CRD42019129167, on the 31st May 2019).

Databases including PubMed/MEDLINE, Science Direct, Cochrane Library, and Google Scholar were systematically searched. Also, we screened the referencelists of identified articles to detect and identify additional relevant studies to add to this review. Furthermore, to find unpublished papers relevant to this systematic review and meta-analysis, Addis Ababa University Digital Library were searched. The search for the literatures was

conductedbetweenthe15[th]of April to the 31[st] of May, 2019. The following terms and keywords were applied for PubMed/MEDLINE search: (infection prevention OR infection control OR standard precaution OR practice) AND (healthcare workers OR health workers OR health personnel OR healthcare providers) AND (health facilities OR hospitals OR public health facilities) AND (Ethiopia) as well as all possible combinations of these terms. For the other electronic databases, we used database-specific subject headings linked with the above terms and keywords used in PubMed. This review is reported according to the Preferred Reporting Items for Systematic Reviews and Meta-Analyses (PRISMA) guidelines [34] (S1 File). The search strategy is provided in supplementary document S2 and S3 Files.

## Inclusion criteria

- Study design: observational studies

- Population: only studies involving healthcare workers

- Language: articles published in the English language

- Reported condition: studies that reported the overall healthcare worker's infection prevention practice

- Availability of full texts

- Study area: studies conducted in Ethiopia

## Exclusion criteria

Articles with the following characteristics were excluded from this review

- Studies whose full data were not accessible even after requests from the authors

- Studies which did not report the overall prevalence of infection prevention practices

- Studies conducted on medical students (1[st] to 4[th] year), health science students, interns, and housekeeping staff

- Qualitative studies, reviews, commentaries, editorials, letters, interventional studies, and other opinion papers

- Excluded published articles with unclear methods

## The outcome of the study

The pooled prevalence of safe infection prevention practices in Ethiopia was the primary outcome variable of this study,a random-effects meta-analysis model was used to estimate the pooled prevalence of safe infection prevention practice. The second objective of this study was to summarize descriptively the factors that were associated with safe infection prevention practices in Ethiopia from the included studies.

## Operational definition

Safe infection prevention practice was defined as healthcare worker's overall compliance to the core components of infection prevention measures that including proper hand hygiene

practice, regular utilization of personal protective equipment's as required, correct medical equipment processing practice, proper healthcare waste management, tuberculosis infection control, and safe injection and medication practices.

## Data extraction

Two investigators (BS and YT) independently extracted the data from the studies included in our analysis as recommended by PRISMA guidelines [34]. The data were extracted using a standard data extraction forms. The following information were extracted from the selected studies: first author's name, year of publication, the type of study design, study setting including region, study population, sample size, sampling methods, the magnitude of infection prevention practice, infection prevention components assessed, and response rateof included studies.

## Quality assessment

The assessment of methodological quality was carried out independently by two reviewers using the Newcastle-Ottawa Scale (NOS) [35]. Thisscale has three sections: $1^{st}$ selection (maximum 5 stars), (2) comparability between groups (maximum 2 stars), and (3) outcome assessment (maximum 3 stars). In summary,the maximum possible score was 10 stars, which represented the highest methodological quality. The two authors (BS and YT) independently assessed the quality of each original study using the quality assessment tool. Any disagreements during the data extraction were resolved through discussion and consensus. Finally, any article with a scale of greater than or equal to $\geq$ 7 out of 10 was included in this Systematic Review and Meta-analysis. A detailed scoring result was described in the supplementary file (S4 File).

## Data analysis and synthesis

Data obtained from the studies under review was entered into Microsoft Excel spreadsheet, then analyzedwere done using STATA Version 14 statistical software. Characteristics of each primary study were presented in a table. The standard errors for each original study were calculated using the binomial distribution formula. The presence of heterogeneity among the reported prevalencewas assessed by computing p-values for the Cochran Q test and $I^2$ test. Cochran's Q test was used to test the null hypothesis of no significant heterogeneity across the studies [36]. Although there can be no absolute rule for when heterogeneity becomes important, Higgins et al. tentatively suggested low for $I^2$ values between 25%–50%, moderate for 50%–75%, and high for $\geq$75% [36]. Subgroup analysis was done by the region where primary studies were conducted, publication year, sample size, sampling method, and type of healthcare facility.

Publication bias was assessed using a funnel plot. In the absence of publication bias, the plot resembles a symmetrical large inverted funnel. Egger's weighted regression and Begg's rank correlation tests were used in checking the publication bias (P < 0.05), considered statistically significant [37]. We also conducted a leave-one-out sensitivity analysis to appraise the main studies that exerted an important impact on between-study heterogeneity.

## Results

### Identification of studies

For this review, one hundred and eighty-seven studies were identified in the initial search. Of these, 118 were excluded during the evaluation of the title and abstract. After applying the inclusion and exclusion criteria, a total of 10 studies were included in the final systematic review and meta-analysis (Fig 1).

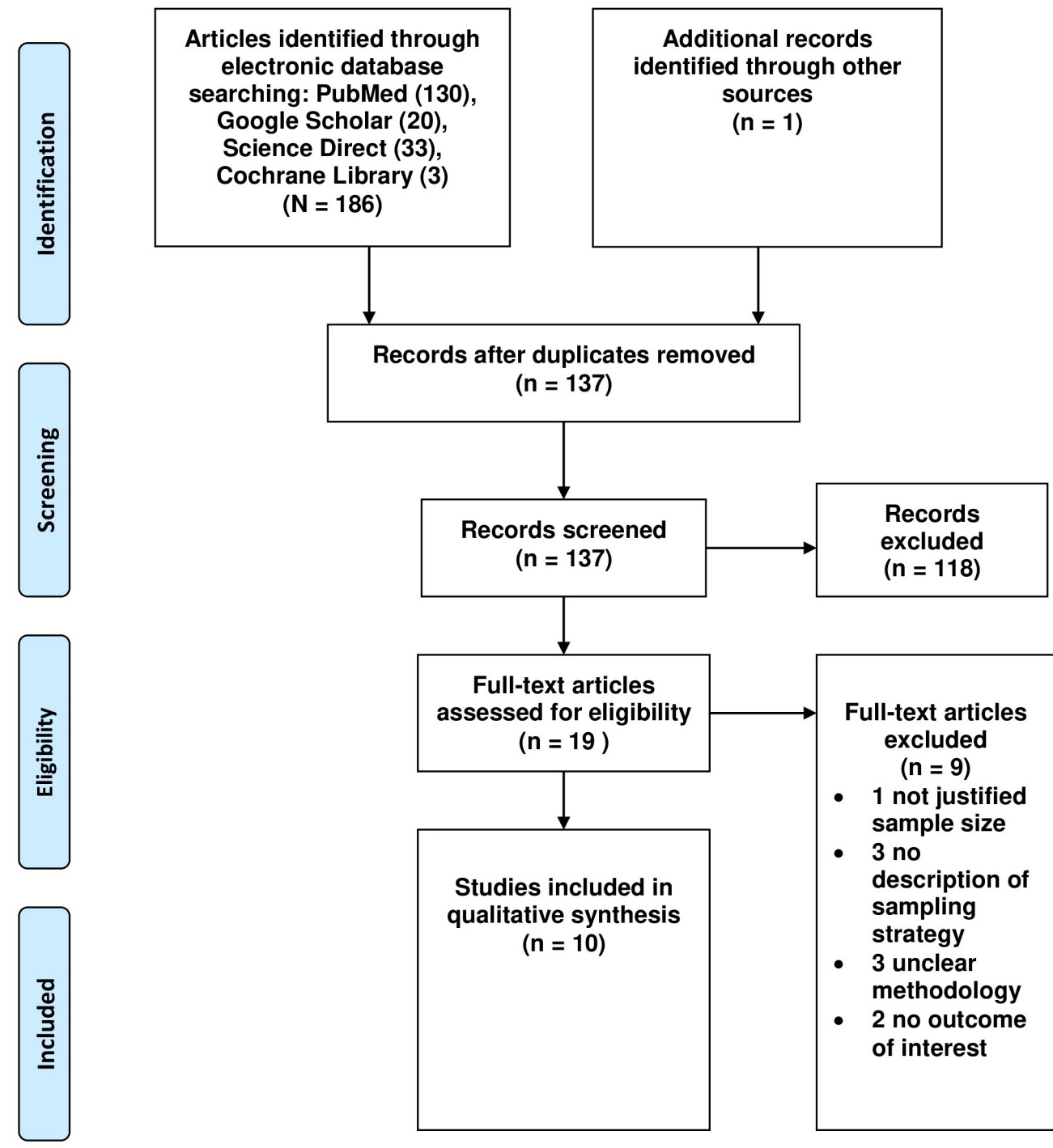

**Fig 1. PRISMA flow chart of review search.**

## Characteristics of included studies

A total of 10 articles [19,20,27,31,33,38–42] were included in meta-analysis. The aggregate study sample included 3,510 participants (a mean of 351 and a median of 314participants). The largest study conducted by Geberemariyam BS., et al had (648 participants) in the Oromia

region [20] while the smallest study by Abreha N., et al. in Addis Ababa had 108 participants [41]. Selected studies were conducted between 2014 and 2019. All the included studies were cross-sectional by design. With regards to regional distribution, about (30%) of the studies were conducted in Addis Ababa [27,40,41]. The prevalence of safe infection prevention practices ranged between 15% [33], and 72.5% [41] in South Nation Nationalities and People (SNNPs) Region and Addis Ababa, respectively. Concerning the quality score, all included studies were of a reputable methodological quality, scoring 7 out of 10-points (Table 1).

## Meta-Analysis

**Prevalence of safe infection prevention practices.** A total of ten studies were included in the meta-analysis. From these studies, the pooled prevalence of safe infection prevention practices in Ethiopia was 52.2% (95%CI: 40.9–63.4). A significant higher heterogeneity among the ten included studies was found ($I^2$ = 98.0%; Q = 453.55, Variance Tau-squared = 319.63, p<0.001). Due to the existence of this heterogeneity, we used a random-effect meta-analysis model to estimate pooled prevalence (Fig 2). According to the sensitivity analysis, there was no single influential study that significantly accounted for it (Table 2).

## Subgroup analyses

**The subgroup analyses of infection prevention practice prevalence.** The results of the subgroup analysis showed that the pooled prevalence of safe infection prevention practices were highest in Addis Ababa (capital city) 66.2% (95%CI: 60.6–71.8) [$I^2$ = 51.4%, p = 0.128], and 54.6% (95%CI: 51.1–58.1) [$I^2$ = 0.0%, p = 0.825] in Amhara Region;48.5% (95%CI: 24.2–72.8) in Oromia Regional State; and the least safe practiceswere reported from other regions (SNNP and Tigray regions)with pooled prevalence of 39.4% (95%CI:13.9–64.8). A considerable heterogeneity was also found [$I^2$ = 97.7%; p<0.001]; and [$I^2$ = 98.8%; p<0.001] for the Oromia Regional State, and other regions (SNNP and Tigray), respectively. The prevalence of infection prevention practices was analyzed separately for either nurses or all other healthcare workers. The findings show the prevalence of safe infection prevention practices more in studies conducted exclusively on nurses than in other health care workers (66.4% vs. 48.6%). We also conducted a subgroup analysis based on the study setting. The pooled prevalence of safe infection prevention practice showed more in studies conducted exclusively in hospitals than in those that include health centers (53.5% vs. 49.8%). More details on the prevalence of safe infection prevention practices for subgroups are presented in Table 3.

## Publication bias

In the present study, Begg's and Egger's tests were utilized to detect the presence of publication bias. However, none of the tests revealed significant publication bias (p-values of 0.210 and 0.246, respectively) for the prevalence of safe infection prevention practice in Ethiopia (Fig 3).

## Sensitivity analysis

Table 2 shows the sensitivity analysis of prevalence for each study being removed at a time. To identify the potential source of heterogeneity in the analysis, a leave-one-out sensitivity analysis on the prevalence of infection prevention practice in Ethiopia was employed. The results of this sensitivity analysis showed that the findings were robust and not dependent on a single study. The pooled estimated prevalence of infection prevention practice varied between 56.2 (95%CI: 48.1–64.4) and 50.0 (95%CI: 38.3–61.6) after removing a single study.

**Table 1. Studies included that shows the prevalence of safe infection prevention practice among healthcare workers in Ethiopia, 2014–2019.**

| Primary author (year) (reference number) | Region location | Study design | Setting | Study population | Sampling | Infection prevention component assessed | Response rate (%) | Sample size | Prevalence with 95%CI | Quality score |
|---|---|---|---|---|---|---|---|---|---|---|
| Sahiledengle B, et al (2018) [27] | Addis Ababa | CS | Hospital & health centers | Nurses, Midwives, Health officers, Physicians, Laboratory technicians, Anesthesiologist, Dentist, Ophthalmologist | Systematic random sampling | Hand hygiene, use of personal protective equipment (PPE), instrument processing, waste management, post-exposure prophylaxis (PEP), TB-infection control, safe injection, and medication practice | 96.2% | 605 | 66.1(64.1–68.0) | 7 |
| Geberemariyam BS., et al (2018) [20] | Oromiya | CS | Hospital & health centers | Physicians, health officers, nurses, midwives, anesthetist, laboratory technicians, pharmacists, environmental health officers, radiographer | Random sampling (Lottery methods) | Hand hygiene, PPE utilization, instrument processing, healthcare waste handling, safe injection | 95.3% | 648 | 36.3(34.4–38.1) | 7 |
| Hussen SH., et al (2017) [38] | SNNP | CS | Referral hospital | Physicians, nurses, laboratory technicians, pharmacist, radiologist | Census | Hand hygiene and healthcare waste handling | 96.7% | 271 | 60.5(57.5–63.4) | 7 |
| Bekele I., et al. (2018) [39] | Oromiya | CS | University hospital | Nurses | Systematic sampling technique | Hand hygiene, PPE utilization, and sharp waste management | 100% | 231 | 61.1(57.8–64.3) | 7 |
| Yohannes T., et al. (2019) [33] | SNNP | CS | General & district hospital | Physicians, nurses, midwives, laboratory technicians, anesthetists, health officers, emergency medical surgeons, specialists, radiographer | Simple random sampling technique | Hand hygiene, use of PPE, instrument processing, waste management | 98.2% | 274 | 15.0(12.8–17.1) | 7 |
| Yallew WW., et al. (2015) [42] | Amhara | CS | Teaching hospital | Physicians, nurse, health officers, health assistants | Systematic random sampling | PPE, blood-borne disease practice, urinary catheter and surgical wound and intravenous catheters | 97.8% | 413 | 55.0(52.5–57.4) | 7 |
| Gebresilassie., et al (2014) [19] | Tigray | CS | Hospital & health centers | Physicians, nurses, midwives, laboratory technicians | Simple random sampling | PPE, hand washing, injection safety | 95.6% | 483 | 42.9(40.6–45.1) | 7 |
| Gulilat K., et al (2014) [31] | Amhara | CS | Hospital, health centers, and private clinic | Physicians, nurses, midwives, laboratory technicians, health officers, sanitarian | Simple random sampling | Hand hygiene, use of PPE, Injection safety | 97.8% | 354 | 54.2(51.5–56.8) | 7 |
| Asmr Y., et al (2019) [40] | Addis Ababa | CS | Specialized and referral hospital | Physicians, nurses | Simple random sampling | Hand washing, PPE, instrument decontamination, | 96.1% | 123 | 60.0(55.5–64.4) | 7 |
| Abreha N., et al. (2018) [41] | Addis Ababa | CS | Specialized hospital | Nurses | Census | Hand washing, PPE, instrument decontamination, waste segregation, PEP | 90.7% | 108 | 72.5(68.2–76.8) | 7 |

CS: Cross-Sectional study, CI: Confidence Interval.

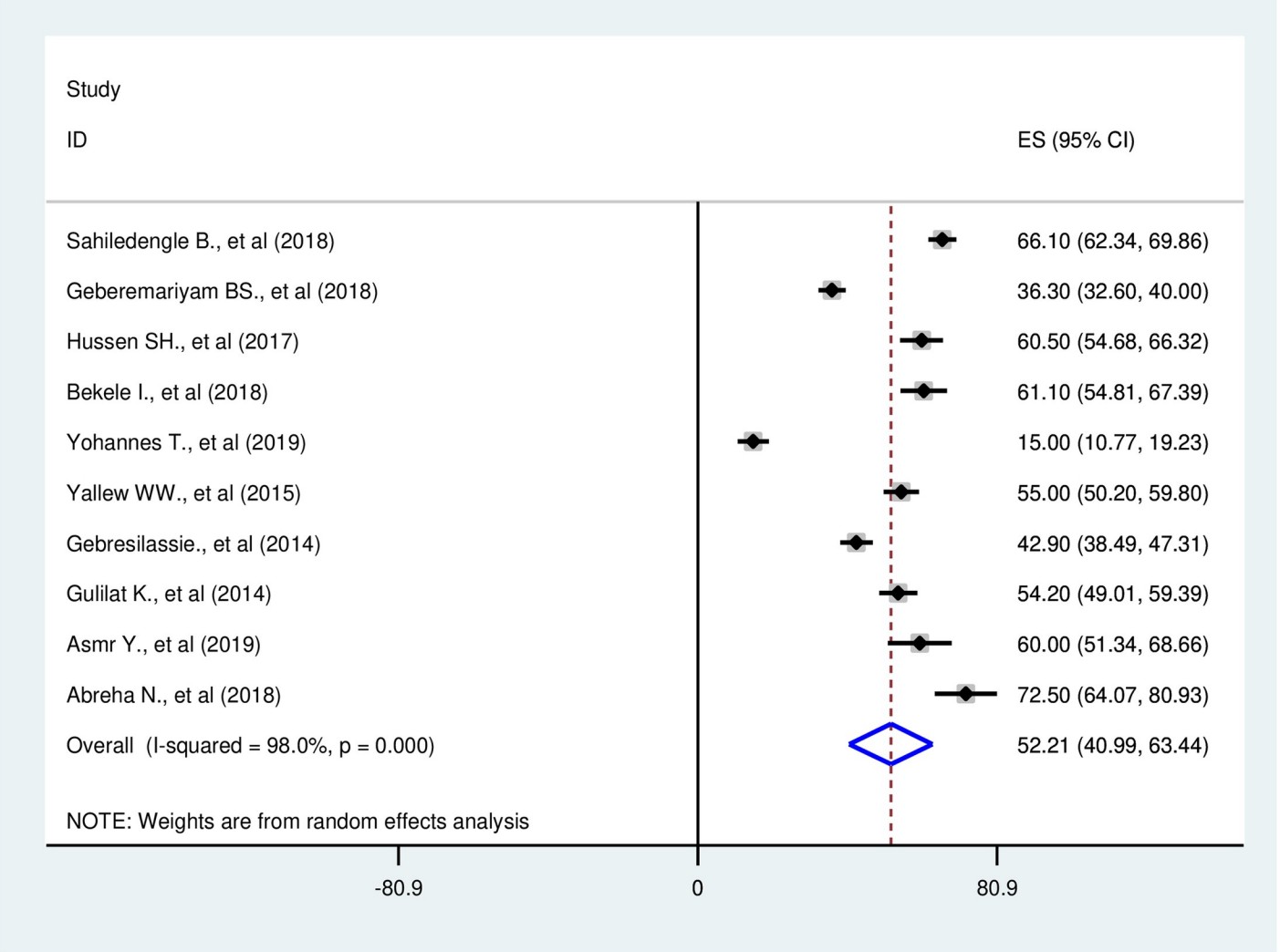

**Fig 2. Forest plot of the pooled prevalence of safe infection prevention practice in Ethiopia, 2014–2019.**

**Table 2. Sensitivity analysis of prevalence for each study being removed at a time: Prevalence and 95% confidence interval of infection prevention practice in Ethiopia, 2014–2019.**

| Study excluded | Prevalence | 95% CI | $I^2$ (%) | Q | p-value |
|---|---|---|---|---|---|
| Sahiledengle B., et al (2018) [27] | 50.6 | 39.1–62.1 | 97.7 | 348.0 | p<0.001 |
| Geberemariyam BS., et al (2018) [20] | 54.0 | 41.6–66.4 | 98.0 | 405.8 | p<0.001 |
| Hussen SH., et al (2017) [38] | 51.3 | 39.1–63.4 | 98.2 | 434.9 | p<0.001 |
| Bekele I., et al (2018) [39] | 51.2 | 39.1–63.3 | 98.2 | 436.2 | p<0.001 |
| Yohannes T., et al (2019) [33] | 56.2 | 48.1–64.4 | 95.6 | 181.3 | p<0.001 |
| Yallew WW., et al (2015) [42] | 51.9 | 39.4–64.4 | 98.2 | 444.8 | p<0.001 |
| Gebresilassie., et al (2014) [19] | 53.2 | 40.5–65.9 | 98.2 | 447.3 | p<0.001 |
| Gulilat K., et al (2014) [31] | 52.0 | 39.5–64.4 | 98.2 | 447.8 | p<0.001 |
| Asmr Y., et al (2019) [40] | 51.3 | 39.4–63.3 | 98.2 | 446.1 | p<0.001 |
| Abreha N., et al (2018) [41] | 50.0 | 38.3–61.6 | 98.1 | 420.3 | p<0.001 |

**Table 3. The subgroup prevalence of safe infection prevention practice in Ethiopia, 2014–2019.**

| Variables | Subgroup | Number of studies included | Sample size | Prevalence (95% CI) | Heterogeneity across the studies | | Heterogeneity between group (p-value) |
|---|---|---|---|---|---|---|---|
| | | | | | $I^2$ (%) | P-value | |
| Region | Addis Ababa | 3 | 836 | 66.2(60.6–71.8) | 51.4 | 0.128 | 0.260 |
| | Oromia | 2 | 879 | 48.5(24.2–72.8) | 97.7 | p<0.001 | |
| | Amhara | 2 | 767 | 54.6(51.1–58.1) | 0.0 | 0.825 | |
| | Others (SNNP & Tigray) | 3 | 1,028 | 39.4(13.9–64.8) | 98.8 | p<0.001 | |
| Type of healthcare facility | Hospital & health centers | 4 | 2090 | 49.8(35.7–63.9) | 97.8 | p<0.001 | 0.741 |
| | Hospital only | 6 | 1420 | 53.9(34.7–73.0) | 98.4 | p<0.001 | |
| Sample size | ≤ 300 | 5 | 1007 | 53.7(29.5–77.8) | 98.0 | p<0.001 | 0.820 |
| | >300 | 5 | 2503 | 50.8(39.5–62.1) | 97.1 | p<0.001 | |
| Profession | Nurses only | 2 | 339 | 66.4(55.2–77.5) | 98.0 | p<0.001 | 0.196 |
| | All type of healthcare workers | 8 | 3171 | 48.6(36.1–61.2) | 98.3 | p<0.001 | |
| Sampling method | Random | 8 | 3131 | 48.7(36.1–61.3) | 98.2 | p<0.001 | 0.206 |
| | Census | 2 | 379 | 66.1(54.3–77.8) | 81.0 | 0.022 | |
| Number of healthcare facilities assessed | ≤ 10 | 7 | 1903 | 52.2(36.9–67.3) | 98.1 | p<0.001 | 0.998 |
| | >10 | 3 | 1607 | 52.1(33.3–71.0) | 98.4 | p<0.001 | |
| Publication year | ≤ 2015 | 3 | 1250 | 50.6(42.6–58.6) | 88.0 | p<0.001 | 0.864 |
| | >2015 | 7 | 2260 | 52.9(36.3–69.5) | 94.7 | p<0.001 | |

SNNP = South Nation Nationalities and Peoples.

Moreover, to identify the possible sources of variations across studies,the meta-regression model was performed by considering the geographical region, publication year, and sample size as covariates. The geographical region (p-value = 0.260), publication year (p-value = 0.864), and sample size (p-value = 0.820) were not statistically significant source of heterogeneity (Table 3).

## Narrative review

From the ten studies, we summarized descriptively the factors that were associated with safe infection prevention practices in Ethiopia. Factors were categorized into the following three domains: socio-demographic factors (four factors), behavior-related factors (three factors), and healthcare facility-related factors (five factors). The overview of these factors including the strength of association and corresponding articles was presented in Table 4.

## Socio-demographic factors

Four socio-demographic factors were significantly associated with safe infection prevention practices. Healthcare workers age [19,41], gender [33,38], profession [19,20,27,42], and higher service year [31] were identified as underlying factors associated with safe infection prevention practice.The odds of safe infection prevention practiceswere higher among the age groupsbetween 20–29 [19], 30–39 [19], and 31–40 [41] than HCWs of greater age. The odds of safe infection prevention practiceswere also higher in female HCWs than males [33,38]. Lastly, significantly lower odds onsafe infection prevention practices were observed amongall

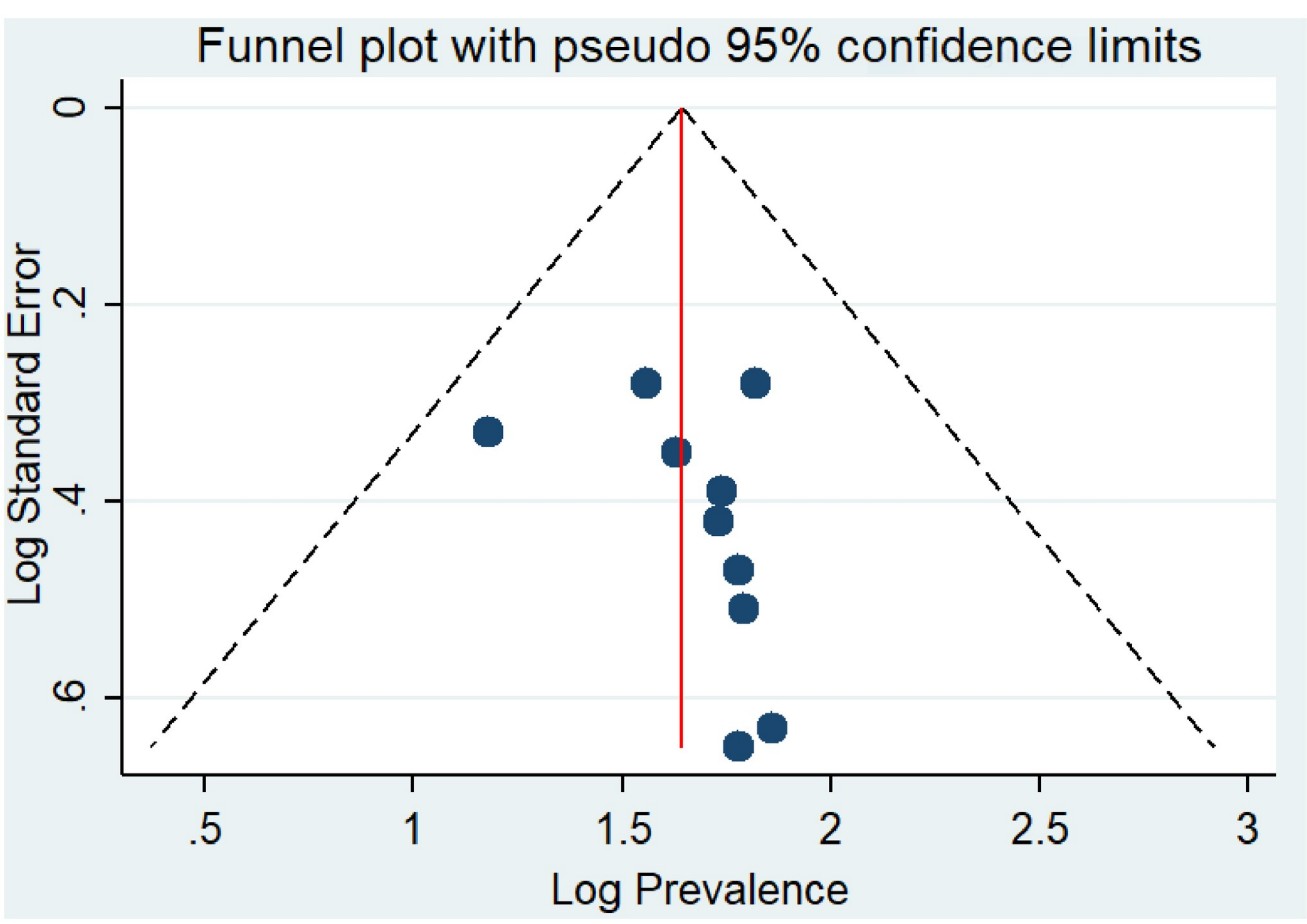

**Fig 3. Funnel plot showing publication bias on prevalence studies among healthcare workers in Ethiopian, a systematic review and meta-analysis, Ethiopia.**

professionals such as midwives [20], laboratory technicians [27], health officers and health assistants [42], and physicians and nurses [19] (Table 4).

### Behavioral related factors

Having good knowledge of infection prevention measures was identified as a factor associated with safe infection prevention practices [27]. In the same way, having a positive attitude towards infection prevention measures, and awareness on infection prevention guideline were the most commonly identified factors associated with the aforementioned practice [27,33] (Table 4).

### Healthcare facility related factors

As illustrated in Table 4, four healthcare facility-related factors were positively and significantly associated with safe infection prevention practices in Ethiopia. Healthcare workers who worked in facilities with continuous water supply have higher odds on safe infection prevention practice [27]. Similarly, healthcare workers who worked in facilities with access to infection prevention guidelines in the working department have higher odds on the prevention practice [19,20,27]. Lastly, factors such as the type of healthcare facility, current working department,

**Table 4. Summary of factors associated with healthcare worker's safe infection prevention practice of studies included in Ethiopia, systematic review, 2014–2019.**

| Author, year | Prevalence of safe infection prevention practice | Variables associated with the univariate analysis | Multivariate analysis factors | Adjusted Odds ratio (AOR) | 95% Confidence interval (CI) | Strength of association |
|---|---|---|---|---|---|---|
| Sahiledengle B., et al (2018) [27] | 66.1% | Current working department: Gynecology, Obstetric, Delivery, OR and Minor-OR Profession: Laboratory technicians and others (Anesthesiologist, Dentist, and Ophthalmologist) Awareness of infection prevention guideline availability Awareness of infection prevention components Presence of hand washing facility Presence of continuous water supply Availability of personal protective equipment Awareness of availability post-exposure prophylaxis available daily/weekly Knowledge of HCWs on infection prevention measures: good Attitude towards infection prevention measures: positive | Profession: Laboratory technicians and others (Anesthesiologist, Dentist, and Ophthalmologist) | 0.18 | 0.07–0.46 | Strong, negative |
| | | | Awareness of infection prevention guideline availability | 1.97 | 1.34–2.93 | Moderate, positive |
| | | | Presence of continuous water supply | 1.68 | 1.11–2.56 | Moderate, positive |
| | | | Good knowledge of infection prevention measures | 1.53 | 1.05–2.22 | Moderate, positive |
| | | | Positive attitude towards infection prevention measures | 2.03 | 1.26–3.26 | Moderate, positive |
| Geberemariyam BS., et al (2018) [20] | 36.3% | Gender: male Profession: midwives Year of service: 10–14 years Availability of water in the working department Presence of infection prevention committee Availability of infection prevention guidelines in the working department Ever taken infection prevention training | Profession: midwives | 0.28 | 0.12–0.69 | Strong, negative |
| | | | Availability of infection prevention guidelines in the working department | 3.34 | 1.65–6.76 | Strong, positive |
| | | | Ever taken infection prevention training | 5.31 | 2.42–11.63 | Strong, positive |
| Hussen SH., et al (2017) [38] | 60.5% | Gender:male Department: Surgical, Obstetrics and gynecology, pediatrics Receiving formal training: no | Gender: male | 0.37 | 0.19–0.74 | Strong, negative |
| | | | Department: Surgical | 0.07 | 0.02–0.20 | Strong, negative |
| | | | Department: Pediatrics | 0.17 | 0.06–0.48 | Strong, negative |
| | | | Receiving formal training | 9.68 | 1.49–81.6 | Strong, positive |
| Bekele I., et al (2018) [39] | 61.1% | Age of healthcare workers (in years) Gender Experience | | | | |
| Yohannes T., et al (2019) [33] | 15.0% | Gender Marital status Work experience The attitude of the respondents towards infection prevention Guidelines Availability of personal protective equipment Accessibility of personal protective equipment Management support for safety Training on infection prevention guidelines | Gender: Female | 2.96 | 1.34–6.53 | Moderate, positive |
| | | | Attitude towards infection prevention guidelines: positive | 3.13 | 1.19–8.22 | Strong, positive |
| | | | Access to infection prevention guidelines | 2.82 | 1.07–7.38 | Moderate, positive |
| | | | Training on infection prevention guidelines | 2.26 | 1.00–5.07 | Moderate, positive |
| Yallew WW., et al (2015) [42] | 55.0% | Profession: Nurses, Health officer, and health assistance | Profession: Nurses | 2.09 | 1.27–3.43 | Moderate, positive |
| | | | Profession: Health officer and health assistance | 0.31 | 0.11–0.84 | Strong, negative |

*(Continued)*

**Table 4.** (*Continued*)

| Author, year | Prevalence of safe infection prevention practice | Variables associated with the univariate analysis | Multivariate analysis factors | Adjusted Odds ratio (AOR) | 95% Confidence interval (CI) | Strength of association |
|---|---|---|---|---|---|---|
| Gebresilassie., et al (2014) [19] | 42.9% | Age of healthcare workers (in years): 20–29, 30–39 Service year of healthcare workers: 1–10 Gender: Male Profession: Doctor, Nurse Presence of written material Training | Age of healthcare workers (in years): 20–29 | 2.6 | 1.1–6.4 | Moderate, positive |
| | | | Age of healthcare workers (in years): 30–39 | 2.5 | 1.1–5.3 | Moderate, positive |
| | | | Gender: Male | 0.5 | 0.3–0.8 | Strong, negative |
| | | | Profession: Doctor | 0.2 | 0.1–0.6 | Strong, negative |
| | | | Profession: Nurses | 0.3 | 0.2–0.6 | Strong, negative |
| | | | Presence of written material | 1.8 | 1.2–2.8 | Moderate, positive |
| | | | Training | 1.6 | 1.0–2.4 | Moderate, positive |
| Gulilat K., et al (2014) [31] | 54.2% | Level of health institution: general and private hospital Availability of personal protective equipment Levels of a profession: Physician Service year Availability of safety box | Level of health institution: general hospital | 2.54 | 1.12–5.75 | Moderate, positive |
| | | | Level of health institution: private hospital | 5.87 | 2.00–17.25 | Strong, positive |
| | | | Availability of personal protective equipment | 6.99 | 2.83–17.27 | Strong, positive |
| | | | Service year: < 10years | 3.79 | 2.33–6.17 | Strong, positive |
| Asmr Y., et al (2019) [40] | 60.0% | Training Profession Infection control guideline in the emergency room Discarded used material as per standard precaution guideline Reused needle or syringe Wash hands before touching the patients Wearing personal protective equipment's before touching the patients Have you ever had Nosocomial infections | | | | |
| Abreha N., et al (2018) [41] | 72.5% | Age of healthcare workers (in years): 31–40 Knowing about infection prevention committee Training on infection prevention | Age of healthcare workers (in years): 31–40 | 3.13 | 1.35–7.25 | Strong, positive |
| | | | Knowing about infection prevention committee | 3.60 | 1.18–10.95 | Strong, positive |
| | | | Training on infection prevention | 2.66 | 1.66–7.37 | Moderate, positive |

and completion of formal infection prevention training, were the most important factors associated with this prevention practice [20,27,33,38,42].

## Discussion

Infection prevention and patient safety in healthcare settings is a nationwide initiative in Ethiopia, that involves the regular implementation of recommended infection prevention practices in every aspect of patient care. Such practices include hand hygiene, injection safety and medication safety, and health care waste management, among others. In Ethiopia, findings regarding the prevalence of safe infection prevention practices have been highly variable. We conducted this systematic review and meta-analysis to estimate the pooled prevalence of safe infection prevention practicesamong HCWs in Ethiopia. Based on the meta-analysis result, only one-half of the HCWs in Ethiopia had safe infection prevention practices. In our

qualitative syntheses, healthcare workers' socio-demographic, behavioral and healthcare facility-related factors were important variables associated with infection prevention practice.

The result of the ten included studies noted that the pooled prevalence of safe infection prevention practice in Ethiopia was 52.2%. This finding brought important information, and these signified that unsafe practices in healthcare facilities are a major public health concern in Ethiopia. As the burdens of HAIs are increasing [14–18], the current suboptimal infection prevention practices have serious implications to both the HCWs and patients.

On one hand, contracting an infection while in the healthcare facilitydue to poor infection prevention practice violates the basic idea that healthcare is meant to make people well. In fact, the risk of contracting HAIs is variable and multifaceted:prevalently, it depends on a patient's immune status, the local prevalence of various pathogens, and the institutional and individual HCW infection prevention practices. Hence, the need for having strong infection prevention programs nationally; and at the healthcare facility level has been not overlooked [29,30,32,43,44]. Un sustained compliance with infection prevention possibly places HCWs at equal, if not at higher risk ofcontracting bacterial and viral infections such as HIV, HBV, HCV, and MRSA in healthcare facilities [9]. In light of this, studies conducted in Ethiopia even showed a positive correlation between poor standard precaution practices and a high prevalence of blood and body fluid exposure [20,27,45,46]. For this reason, the Federal Ministry of Health infection control professionals, healthcare facility administrators, and hospital epidemiologists must pay considerable attention to curve the current poor suboptimal infection prevention practices [47,48].

In the subgroup analysis, a variation in HCWs infection prevention practices across geographical regions was found. Safe infection prevention practices were consistently more frequent in central Ethiopia (Addis Ababa) and less in Tigray and SNNP regions-the reason for these regional differences may be explained by studies conducted in central Ethiopia included mainly in tertiary and referral hospitals which and commonly staffed are with skilled and experienced healthcare professionals as compared to those in other regions. Another possible explanation for this variation might be due to the difference in environmental infrastructures and behavioral characteristics of HCWs. Our findings may, therefore, indicate the need to promote appropriate infection prevention and patient safety practices for HCWs in Ethiopia. Moreover, to address regional variationsthere is a strong need of implementing readily available, relatively inexpensive, practical and scientifically proven infection prevention and patient safety practices in different regions of Ethiopia.

Our meta-analysis also found that the prevalence of safe infection prevention practices differed between nurses and other healthcare workers. The possible explanation for this observed discrepancy may be due to the training and roles of healthcare workers; the nurses were engaged in inpatient care, and they may have better understanding regarding infection prevention. Still, this prevalence is suboptimal and great concern, therefore, is necessary to strive for a better quality of healthcare.

In this review,we summarize the findings of the included studies on factors associated with safe infection prevention practice identified three main domains of determinant factors; namely socio-demographic, behavioral, and healthcare facility-related factors. Healthcare workers in facilities with access to infection prevention guidelines and those receiving formal infection prevention training have higher odds onsafe infection prevention practice. Obviously, this may be due to health professionals who have adequate knowledge and attitude to implement the recommended infection prevention and patient safety practices in the healthcare facilities possibly have better IPC compliance [27]. In this sense, the current systematic review suggests that it may be more effective to improve HCWs infection prevention practices through regular in-service training [49]. Furthermore, a holistic approach that involves the

behaviors of HCWs and facilities that are essential for effective infection prevention and control measures should be integrated. Since infection Prevention and Patient Safety recommendations could easily be implemented if everyone in the health service delivery system, from the level of policy makers to healthcare providers at the facility level collaborate [9,27,31].

Finally, despite there were similar trends for many of the African countries in the practice of healthcare worker's infection prevention and control practice, we would suggest caution against applying the present results to countries located in other regions of the African, as the healthcare system, healthcare workers training, and a government policy may affect HCWs infection prevention compliance.

## Limitations of the study

This systematic review and meta-analysis have several limitations. The first limitation considered to conduct this review was to include English language articles only. Second, all of the studies included in this review were cross-sectional as a result; the outcome variable might be affected by other confounding variables. Third, this meta-analysis represented only studies that were reported from the four regions of the country- this irregular distribution of studies from around the country limits the study findings. Fourth, the majority of the studies included in this review had relatively small sample sizes which could have affected the estimated safe infection prevention practice reports. Fifth, a small number of studies were included in subgroup analyses which reduce the precision of the estimate and considerable heterogeneity was identified among the studies. Sixth, almost all studies included in this meta-analysis were often based on self-reported data from healthcare providers, which tended to have overestimated compliance and limited the strengths of the findings. Lastly, since most of the included primary studies did not cover a good range of components of infection prevention practices. We strongly recommend caution while interpreting the estimated pooled prevalence finding.

## Conclusions

Infection prevention practices in Ethiopia was poor, with only half of the healthcare workers reportingsafe practices. There were regional and professional variations in the prevalence on the safe practices-it is therefore important for all HCWs to adhere to the existing infection control guidelines by embedding them in everyday practice. It is also imperative for healthcare administrators to ensure the implementation of infection prevention and patient safety programs in all healthcare settings. Our study highlights the need for the Ethiopian Federal Ministry of Health to step-up efforts to intensify the current national infection prevention and patient safety initiatives.

## Supporting information

**S1 File. PRISMA checklist.**
(DOC)

**S2 File. Search strategy (full searching strategies for PubMed).**
(DOCX)

**S3 File. Search strategy (example Google scholar).**
(DOCX)

**S4 File. Methodology quality assessment of included and excluded studies.**
(DOCX)

## Acknowledgments

The authors would like to thank MaddaWalabu University Goba Referral Hospital Public Health Department staff for providing their unreserved support. We would like to thank for the valuable support we received from Mr. John Edward Quisido (assistant professor)as well as Dr. David Allison for their proofreading support.

## Author Contributions

**Conceptualization:** Biniyam Sahiledengle.

**Data curation:** Biniyam Sahiledengle.

**Formal analysis:** Biniyam Sahiledengle, Yohannes Tekalegn, Demelash Woldeyohannes.

**Investigation:** Biniyam Sahiledengle, Yohannes Tekalegn, Demelash Woldeyohannes.

**Methodology:** Biniyam Sahiledengle, Yohannes Tekalegn, Demelash Woldeyohannes.

**Project administration:** Biniyam Sahiledengle.

**Supervision:** Biniyam Sahiledengle, Yohannes Tekalegn.

**Validation:** Biniyam Sahiledengle, Yohannes Tekalegn, Demelash Woldeyohannes.

**Visualization:** Biniyam Sahiledengle.

**Writing – original draft:** Biniyam Sahiledengle.

**Writing – review & editing:** Biniyam Sahiledengle, Yohannes Tekalegn, Demelash Woldeyohannes.

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
