## [Decision Letter · Decision Letter 0]

30 Oct 2020

PONE-D-20-11122

The critical role of infection prevention overlooked in Ethiopia, only one-half of health-care workers had safe practice: A Systematic Review and Meta-Analysis

PLOS ONE

Dear Dr. Sahiledengle,

Thank you for submitting your manuscript to PLOS ONE. After careful consideration, we feel that it has merit but does not fully meet PLOS ONE’s publication criteria as it currently stands. Therefore, we invite you to submit a revised version of the manuscript that addresses the points raised during the review process.

We look forward to receiving your revised manuscript.

Kind regards,

Holly Seale

Academic Editor

PLOS ONE

Journal Requirements:

2. We note that your literature search was performed on May 2019;to allow an up-to-date view of the topic, we would request that the search is updated. Moreover, please report in more detail the results of the quality assessment, showing how each included study scored in every item of the scale.

Reviewers' comments:

Reviewer's Responses to Questions

**Comments to the Author**

1. Is the manuscript technically sound, and do the data support the conclusions?

Reviewer #1: Partly

Reviewer #2: Yes

2. Has the statistical analysis been performed appropriately and rigorously? 

Reviewer #1: Yes

Reviewer #2: Yes

3. Have the authors made all data underlying the findings in their manuscript fully available?

Reviewer #1: Yes

Reviewer #2: Yes

4. Is the manuscript presented in an intelligible fashion and written in standard English?

Reviewer #1: No

Reviewer #2: Yes

5. Review Comments to the Author

Reviewer #1: The topic chosen for the research paper is appropriate and interesting, however the background of the study lacks cohesion. Although, the beginning of the background section is informative however the problem defined has not been comprehensively supported by relevant research articles. The focus of the background section is on infection prevention of patients rather than on healthcare workers about whom the research paper is about. The methodology section needs more work for e.g. the author should explain what does ‘medical’ signify in the exclusion criteria and who are healthcare workers. The outcome of the study in the methodology section should be more comprehensive to begin with. In the discussion section the author should elaborate more at places where claims are made about poor infection control among healthcare workers. The claims should be supported by examples and the author should check for grammatical mistakes and typo errors for example.

Line 12: ‘such as such’ Typo.

Line 14 ‘maximize patient outcomes’: What the authors mean by maximize here is not clear.

Line 23 ‘Cochrane Q test’ : It should be Cochran Q test .

Line 51: what is HAIs?

Line 72 : ‘Ethiopia, similar to other African countries, does not have a well-described report on the burden of ‘: The reader would like to know which are the other African countries.

Line 79 : ‘Sufficient evidence that demonstrated the role of infection prevention on the reduction of HAIs’ : The reader would like to know about whom the authors are addressing, patients/healthcare workers?

Line 84: ‘policies, and technical guidelines made the problem even worse’ : The reader would like to know about how did healthcare workers attitude made the situation worse.

Line 87 : ‘the publication of the second 87 national infection prevention and patient safety guidelines was released.’ This statement does not convey the message and may confuse the reader.

Line 88: ‘ From that day on, 88 considerable progress has been made in understanding the basic principles, acceptance, and use of 89 evidence-based infection prevention practices in Ethiopia’ : The reader would like to know more about the progress

Line 90: ‘reported inconsistent findings’ : The reader would like to know more about inconsistencies.

Line 139: ‘ Studies conducted on medical’ : What does medical signify here. Needs a definition.

Line 301: ‘recommended infection prevention principles among HCWs in developing

countries is poor’ : This can mislead the reader needs more detailed explanation.

Line 335: ‘systematic intervention measures’ : The reader would like to know more about these interventions.

Line 347: ‘prevention knowledge, might increase compliance’: This statement is not clear.

Line 349: ‘holistic approach’: The authors have not discussed what do they understand by holistic approaches anywhere in the document.

Reviewer #2: Ref: PONE-D-20-11122

The critical role of infection prevention overlooked in Ethiopia, only one-half of health-care workers had safe practice: A Systematic Review and Meta-Analysis

Dear Editor

Thank for you for the opportunity to review this manuscript. The study reviews the prevalence of safe prevention practices and the associated factors among healthcare workers in Ethiopia.

General comments:

This study is of interest to the readers, particularly those seeking to understand the situation of safe infection prevention in Ethiopia and its drivers.

The authors calculated the pooled prevalence from the prevalence of safe infection prevention practices reported by healthcare workers. In my opinion, the method would more perfect in estimating the awareness/knowledge of health care workers rather their practices.

What is the difference between “safe infection prevention” and “infection prevention” throughout the document? In some context, the words bring confusion than clarity.

Some repetitive phrases, grammatical error and typos to correct throughout the document need to be addressed eg page 8Ln12, pg10Ln51, pg10Ln66, 75, 79, 232

Specific areas:

Abstract

In which direction these factors were associated with safe prevention practices? Were they risk or protective factors?

Background

The authors highlighted the release of the second edition of National infection prevention publication in 2012. What are the proposed components of infection prevention practices as per this document in the country and how do they relate to the components assessed by the authors?

Methodology

The authors reported 10 studies qualified for systematic review and meta-analysis based on inclusion and exclusion criteria. Since being both a quantitative and qualitative was not an inclusion criteria; I failed to figure out the number of studies reviewed per study design: quantitative and qualitative. Were all qualified studies both quantitative and qualitative?

Outcome of the study

The definition of the primary outcome and how it was computed don't match, the computation gives an impression of the awareness/knowledge of the practices and not how they practice.

Results

Table 1: what were the components assessed by for the Bekele I et al, Hussen SH., et al studies?

Table 3: What is the reasoning behind the cut-off points in sample size and publication year variables? (Was there a new practise introduced before or after 2015?)

Discussion

How does this pooled prevalence compare to other findings elsewhere in Africa, for example.

Limitation

Most of the studies reviewed by the authors did not cover a good range of components of infection prevention practices. Could this be a limitation to keep in mind when presenting the obtained pooled prevalence?

The authors did not mitigate any of the limitations of the study

References

Include access date for web-based references.e.g. page32Ln391

6. PLOS authors have the option to publish the peer review history of their article (what does this mean?). If published, this will include your full peer review and any attached files.

Reviewer #1: **Yes: **Mohammed Owais Qureshi

Reviewer #2: **Yes: **Erick Kinyenje

---

## [Author Response · Author response to Decision Letter 0]

30 Nov 2020

Author response to reviewer’s comments

Dear reviewers 

Thank you for this learning opportunity. We are so glad to see our paper improved because of your comments and wise advice. Please follow a point by point response to the reviewer’s comment. We used a “Yellow Color Text Highlight” for all affected revisions and corrections in the “Revised Manuscript”. Please follow a point by point response.

Thank you for this opportunity

For Reviewer #1

Thank you our respected reviewer #1, 

We would like to appreciate your interesting comments and suggestion. As per your wise advice we addressed all your concerns accordingly. Please follow a point by point response and see the revised manuscript.

Reviewer #1: The topic chosen for the research paper is appropriate and interesting, however the background of the study lacks cohesion. Although, the beginning of the background section is informative however the problem defined has not been comprehensively supported by relevant research articles. The focus of the background section is on infection prevention of patients rather than on healthcare workers about whom the research paper is about. The methodology section needs more work for e.g. the author should explain what does ‘medical’ signify in the exclusion criteria and who are healthcare workers. The outcome of the study in the methodology section should be more comprehensive to begin with. In the discussion section the author should elaborate more at places where claims are made about poor control among healthcare workers. The claims should be supported by examples and the author should check for grammatical mistakes and typo errors for example.

Response

Thank you our respected review for your comment. Following your wise advice, we revised the background section of the manuscript and we try to focus on healthcare worker's infection prevention practices (please see the revised manuscript section background section). Regarding, the exclusion and outcome variables issue we clarified for better understanding in the revised manuscript (please see the revised manuscript method section). Also, we checked for grammatical mistakes and typo errors throughout the manuscript. Thank you.

Line 12: ‘such as infection such’ Typo.

Response

Thank you for your comment. We corrected this and many other typos in the revised manuscript. Please see the revised manuscript file. Thank you.

Line 14 ‘maximize patient outcomes’: What the authors mean by maximize here is not clear.

Thank you for your comment. We clarified this statement in the revised manuscript. Please see the revised manuscript file abstract section. Thank you.

We corrected it as "Background: Effective infection prevention and control measures, such as proper hand hygiene, the use of personal protective equipment, instrument processing, and safe injection practice in the healthcare facilities are essential elements of patient safety and lead to optimal patient outcome. In Ethiopia, findings regarding infection prevention practices among healthcare workers have been highly variable and uncertain. This systematic review and meta-analysis estimate the pooled prevalence of safe infection prevention practices and summarizes the associated factors among healthcare workers in Ethiopia."

Line 23 ‘Cochrane Q test’ : It should be Cochran Q test .

Response:

Thank you for your comment. We corrected this and many other typos in the revised manuscript. Please see the revised manuscript file. Thank you.

Line 51: what is HAIs?

Response:

Thank you for your comment. We fully described what HAIs (Healthcare Acquired Infections) means as the word appeared for the first time in the background section. Thank you for your correction.

Line 72 : ‘Ethiopia, similar to other African countries, does not have a well-described report on the burden of ‘: The reader would like to know which are the other African countries.

Response

Thank you for these interesting comments. As per your wise advice we rephrase and revised the whole paragraph for a better explanation of our idea. Please see the revised manuscript background section. Thank you again.

We rewrite the paragraph as "In Ethiopia, the burden of HAIs are a major public health problem with a significant impact on hospitalized patients [14-16]. According to the finding of some pocket studies a high prevalence of HAIs has been reported from all corners of the country-from 15.4% in north Ethiopia [15], 11.4%-19.4% in southwest Ethiopia [16,17], to 16.4% in central Ethiopia [18]. Although, a large proportion of HAIs can be prevented with inexpensive and cost-effective infection prevention and control measures; the evidence available suggests that healthcare facilities in Ethiopia do not have effective infection control programs [9]. Also, HCWs compliance towards infection prevention and control (IPC) measures are critically low and a potential common problem in the country [9,19, 20]. ”

Line 79: ‘Sufficient evidence that demonstrated the role of infection prevention on the reduction of HAIs’ : The reader would like to know about whom the authors are addressing, patients/healthcare workers?

Response

Thank you for your comment. In short, we try to address healthcare workers. Following your comment, we clarified the paragraph for better clarification. Please see the revised manuscript background section. Thank you again.

We revised the paragraph in the following manner “There is evidence from that demonstrates the role of HCWs infection prevention compliance on the reduction of HAIs [21-23]- for example, Sickbert et al, in their study reported an improvement in hand hygiene compliance of healthcare workers by 10%, which associated a significant reduction in overall HAIs [22].”

Line 84: ‘policies, and technical guidelines made the problem even worse’ : The reader would like to know about how did healthcare workers attitude made the situation worse.

Response

Thank you again for your interesting comments. We rewrite the statement for better clarification. Please see the revised manuscript background section. Thank you again.

We revised the paragraph in the following manner "In this context, adherence to the recommended infection prevention and patient safety practice is the best ways in preventing patients, healthcare workers, and communities at large from HAIs. And the long-term solution to reduce the problems of HAIs lies in actions to implement effective IPC measures in healthcare facilities [3,9,10,13]. Despite these facts-in many low-income settings, with healthcare systems and resources similar to Ethiopia, lack of well-trained HCWs, lack of infection prevention and control policies, and lack of technical guidelines consistent with the available evidence essential to provide a robust framework to support the performance of good IPC practices made the promotion of IPC practices a bit challenging [9,15,24-28]."

Line 87 : ‘the publication of the second 87 national infection prevention and patient safety guidelines was released.’ This statement does not convey the message and may confuse the reader.

Response: 

Thank our respected review. We apologized for not making this very clear. As per your wise advice, we corrected accordingly in the following manner "Since the publication of the second Ethiopia National Infection Prevention Guidelines in 2012 [9], considerable progress has been made in understanding the basic principles, acceptance, and use of evidence based Infection Prevention (IP) practices. The manual serves as a standardized IP reference manual for healthcare providers in all healthcare delivery systems. Also, it is intended to serve HCWs by providing clear guidance in the provisions of standard infection prevention and patient safety practices. The key components in the manual include standard precautions, hand hygiene, personal protective equipment, safe injection practice, processing instrument, and healthcare waste management [9]." Please the revised manuscript background section (line 87-89).

Line 88: ‘ From that day on, 88 considerable progress has been made in understanding the basic principles, acceptance, and use of 89 evidence-based infection prevention practices in Ethiopia’ : The reader would like to know more about the progress

Response: 

Thank you for your comments. As per your wise advice, we rewrite and briefly describe what has been achieved in the area of infection prevention. Please see the revised manuscript background section (lines 86-100).

"To maximize the prevention of HAIs and promote safe IPC measures at the national level in Ethiopia, there has been a growing recognition of the need for safe infection prevention and patient safety practice at all levels. Since the publication of the second Ethiopia National Infection Prevention Guidelines in 2012, considerable progress has been made in understanding the basic principles, acceptance, and use of evidence-based Infection Prevention (IP) practices, including Clean and Safe Hospital (CASH), Clean Care is Safer Care campaigns, and Initiatives-Saving Lives through Safe Surgery (SaLTS). The national Infection Prevention and Patient Safety (IPPS) manual is intended to serve healthcare providers and managers by providing clear guidance in the provisions of standard infection prevention and patient safety practices for healthcare facilities in Ethiopia [9]. Importantly, the existence of IPC guidelines alone is not sufficient to ensure their adoption and implementation and findings indicate that HCWs compliance is a prerequisite for successful guideline adoption. Previously conducted primary studies reported inconsistent findings regarding HCWs infection prevention practice in Ethiopia [19,20,27,29-33]."

Line 90: ‘reported inconsistent findings’ : The reader would like to know more about inconsistencies.

Response: 

Thank you for your comments. As per your wise advice, we describe this inconsistency. Please see the revised manuscript background section (line 99-103).

"Previously conducted primary studies reported inconsistent findings regarding HCWs infection prevention practice in Ethiopia [19,20,27,29-33]. For instance, a study done in southeast Ethiopia showed that only 36.3% of HCWs had safe infection prevention practice [20], 15.0% in southern Ethiopia [33], 66.1% in central Ethiopia [27], and in north Ethiopia, 42.9% of HCWs had acceptable practice [19]."

Line 139: ‘ Studies conducted on medical’ : What does medical signify here. Needs a definition.

Response: 

Thank you for your comment. We apologized for not making this very clear. In short, medical signify, 1st to 4th year medical students. In Ethiopia, first to fourth year medical students are not called interns rather they are called "medical studies” . While final year medical studies commonly called “Interns”. To make this clear for the international audience we paraphrase the statement as “Studies conducted on medical students (1st to 4th year), health science students, interns, and housekeeping staff”

Line 301: ‘recommended infection prevention principles among HCWs in developing

countries is poor’ : This can mislead the reader needs more detailed explanation.

Response: 

Thank you for your comment. As per your wise advice, we paraphrase the stated description "Infection prevention and patient safety in healthcare settings is a nationwide initiative in Ethiopia, that involves the regular implementation of recommended infection prevention practices in every aspect of patient care. Such practices include hand hygiene, injection safety and medication safety, and health care waste management, among others." Please see the revised manuscript.

Line 335: ‘systematic intervention measures’ : The reader would like to know more about these interventions.

Response: 

Thank you for your comment. To make our discussion clear we restate the description as "Our findings may, therefore, indicate the need to promote appropriate infection prevention and patient safety practices for HCWs in Ethiopia. Moreover, to address regional variations there is a strong need of implementing readily available, relatively inexpensive, practical, and scientifically proven infection prevention and patient safety practices in different regions of Ethiopia." Please see the revised manuscript.

Line 347: ‘prevention knowledge, might increase compliance’: This statement is not clear.

Response: 

Thank you for your comment. We again apologize for this proofread error. Following your wise recommendation, we corrected it accordingly. "Obviously, this may be due to health professionals who have adequate knowledge and attitude to implement the recommended infection prevention and patient safety practices in the healthcare facilities possible have better IPC compliance [27]."

Line 349: ‘holistic approach’: The authors have not discussed what do they understand by holistic approaches anywhere in the document.

Response: 

Thank you for your comment. In short, a "holistic approach" signifies a combination of infection prevention control measures–including facility-related issues and healthcare worker's related behavior. As per your wise advice, we clarified the paragraph in the following manner "Furthermore, a holistic approach that involves the behaviors of HCWs and facilities that are essential for effective infection prevention and control measures should be integrated. Since infection Prevention and Patient Safety recommendations could easily be implemented if everyone in the health service delivery system, from the level of policymakers to healthcare providers at the facility level collaborate [9,27,31]." Please see the revised manuscript discussion section.

Thank you with all respect. 

For Reviewer #2

Thank you our respected reviewer #2, 

This is a privilege for us to have your comments. Our respected reviewer, thank you for your advice. All your comments are very important and we address your comments accordingly. Please follow a point by point response and see the revised manuscript.

With all respect. 

Reviewer #2: Ref: PONE-D-20-11122

The critical role of infection prevention overlooked in Ethiopia, only one-half of health-care workers had safe practice: A Systematic Review and Meta-Analysis

Dear EditorThank you for the opportunity to review this manuscript. The study reviews the prevalence of safe prevention practices and the associated factors among healthcare workers in Ethiopia.

General comments:

This study is of interest to the readers, particularly those seeking to understand the situation of safe infection prevention in Ethiopia and its drivers.

The authors calculated the pooled prevalence from the prevalence of safe infection prevention practices reported by healthcare workers. In my opinion, the method would more perfect in estimating the awareness/knowledge of health care workers rather their practices.

What is the difference between “safe infection prevention” and “infection prevention” throughout the document? In some context, the words bring confusion than clarity.

Some repetitive phrases, grammatical error and typos to correct throughout the document need to be addressed eg page 8Ln12, pg10Ln51, pg10Ln66, 75, 79, 232

Response:

Thank you for your comment. In brief, infection prevention and control is a set of practices, protocols, and procedures that are put in place to prevent infections that are associated with the healthcare system. This includes measures such as hand hygiene, personal protective equipment (PPE) utilization, instrument processing, injection, and medication safety, environmental cleaning, healthcare waste management, and standard precaution. Compliance with these recommended measures is termed as safe infection prevention practice. Our respected reviewer to make this thing very clear we describe this issue in the revised manuscript background section (line 41-57).

Specific areas:

Abstract

In which direction these factors were associated with safe prevention practices? Were they risk or protective factors?

Response:

Thank you for your comment. As per your wise advice, we included the direction of the association. Since we qualitatively syntheses factors that are associated with healthcare worker's safe infection prevention practices we did not report the odds ratio (please see table 4). "In our qualitative syntheses, the odds of safe infection prevention practice were higher among healthcare workers who had good knowledge and positive attitude towards infection prevention. Also, healthcare workers working in facilities with continuous running water supply, having infection prevention guideline, and those received training were significantly associated with higher odds of safe infection prevention practice." Please see the revised manuscript abstract section.

Background

The authors highlighted the release of the second edition of National infection prevention publication in 2012. What are the proposed components of infection prevention practices as per this document in the country and how do they relate to the components assessed by the authors?

Response: 

Thank you for your comment. As per your wise advice, we included the proposed components of infection prevention practices as per the IPPS document "The manual serve as a standardized IP reference manual for healthcare providers in all healthcare delivery systems. Also, it is intended to serve HCWs by providing clear guidance in the provisions of standard infection prevention and patient safety practices. The key components in the manual include standard precautions, hand hygiene, personal protective equipment, safe injection practice, processing instrument, and healthcare waste management [9]." Please see the revised manuscript background section.

Methodology

The authors reported 10 studies qualified for systematic review and meta-analysis based on inclusion and exclusion criteria. Since being both a quantitative and qualitative was not an inclusion criteria; I failed to figure out the number of studies reviewed per study design: quantitative and qualitative. Were all qualified studies both quantitative and qualitative?

Response:

Thank you for your comment. As per the preferred reporting items for systematic reviews and meta-analyses /PRISMA 2009 Flow Diagram: Moher D, Liberati A, Tetzlaff J, Altman DG, The PRISMA Group (2009). Preferred Reporting Items for Systematic Reviews and Meta-Analyses: The PRISMA Statement. PLoS Med 6(7): e1000097. doi:10.1371/journal.pmed1000097" . We include 10 studies in our systematic reviews and meta-analyses and all are quantitative studies. As per your advice, we corrected figure 1 accordingly. Please see the revised Figure 1.

Outcome of the study

The definition of the primary outcome and how it was computed don't match, the computation gives an impression of the awareness/knowledge of the practices and not how they practice.

Response:

Thank you for your comments. In brief, a random-effects meta-analysis model was used to estimate the pooled prevalence of safe infection prevention practice for computation of the outcome variable. For a better description, we rewrite the outcome variable and include the operational definition section.

The outcome of the study

The pooled prevalence of safe infection prevention practices in Ethiopia was the primary outcome variable of this study, a random-effects meta-analysis model was used to estimate the pooled prevalence of safe infection prevention practice. The second objective of this study was to summarize descriptively the factors that were associated with safe infection prevention practices in Ethiopia from the included studies.

Operational definition

Safe infection prevention practice was defined as healthcare worker’s overall compliance to the core components of infection prevention measures that including proper hand hygiene practice, regular utilization of personal protective equipment’s as required, correct medical equipment processing practice, proper healthcare waste management, tuberculosis infection control, and safe injection and medication practices. 

Please see the revised manuscript. 

We remove the following statement “The prevalence was computed by dividing the number of healthcare workers reported correct/acceptable infection prevention practices by the total number of healthcare workers (sample size), and multiplied by 100. ” in order to make outcome of the study very clear. Basically, we include this statement to show how the primary studies reported the prevalence of infection prevention practice in their report. studies 

Results

Table 1: what were the components assessed by for the Bekele I et al, Hussen SH., et al studies?

Response: 

Thank you our respected reviewer for your advice. We included the components assessed by Bekele I et al, Hussen SH., et al. Please see the revised table 1. 

Table 3: What is the reasoning behind the cut-off points in sample size and publication year variables? (Was there a new practice introduced before or after 2015?)

Response: 

Thank you our respected reviewer. We conducted a sub-group analysis as a means of investigating the sources of heterogeneity. A subgroup analysis was done considering the publication year and sample size of primary studies. The assumption behind selecting the year 2015 was based on the Ethiopia Federal Ministry of Health (FMOH) Health Sector Transformation Plan (HSTP) as part of the second Growth and Transformation Plan of the Ethiopian government. The HSTP has set targets toward realizing the Sustainable Development Goals and identified transformation agendas. In line with the quality and equity transformation agenda and as part of recognizing the key roles essential and emergency surgical care plays in achieving universal health coverage, the FMOH has prioritized surgical and anesthesia care by launching the national flagship Initiative-Saving Lives through Safe Surgery (SaLTS). The FMOH also implemented several initiatives such as Respectful and Caring (CRC) health workforce, CASH (Clean and Safe Hospital), Infection Prevention and Patient Safety (IPPS), WASH FIT in health care facilities, and many other initiatives.

Regarding the sample size, we used the sample size of 300 as a cut off point for investigating the sources of heterogeneity based on the median value of the primary studies included in our systematic review. We describe the aggregate study sample size as well as the median value (which is 314) in our manuscript (please see the revised manuscript result section line 204-206). 

Discussion

How does this pooled prevalence compare to other findings elsewhere in Africa, for example.

Response:

Thank you for the comment. In the discussion selection we cautious while we discuss the meta-analysis finding; as you understand this is the first systematic review in Ethiopia and probability in Africa also, as a result, it is difficult to compare our systematic review with another related review due to lack of related reviews. Also, as the present study estimated the pooled prevalence comparting the present finding with individual studies is very difficult and problematic, as a result, we restrict our self not to do so. Yet, we discuss our findings in a detailed manner considering our study settings. Finally, we insert the following paragraph at the end of the discussion section.

“Finally, despite there were similar trends for many of the African countries in the practice of healthcare worker’s infection prevention and control practice, we would suggest caution against applying the present results to countries located in other regions of the African, as the healthcare system, healthcare workers training, and a government policy may affect HCWs infection prevention compliance.”

Limitation

Most of the studies reviewed by the authors did not cover a good range of components of infection prevention practices. Could this be a limitation to keep in mind when presenting the obtained pooled prevalence?

Response:

Thank you for this interesting insight and comment. Our respected reviewer you right. We include this issue in the study limitation section. Please see the revised manuscript limitation section.

We included the following statement 

“Lastly, since most of the included primary studies did not cover a good range of components of infection prevention practices. We strongly recommend caution while interpreting the estimated pooled prevalence finding.”

The authors did not mitigate any of the limitations of the study

Response:

Thank you for your comment. As the current systematic review depends on the finding of the primary studies we are unable to mitigate the limitations we mentioned in the limitation of the study section, and almost all limitations were directly related to the findings of the primary studies. Thank you.

References

Include access date for web-based references.e.g. page32 Ln391

Response: 

Thank you for the comments. As per your advice, we include the access date for all web-based references. Please see the revised manuscript reference section. 

Thank you with all respect.

---

## [Decision Letter · Decision Letter 1]

4 Jan 2021

The critical role of infection prevention overlooked in Ethiopia, only one-half of health-care workers had safe practice: A Systematic Review and Meta-Analysis

PONE-D-20-11122R1

Dear Dr. Sahiledengle,

We’re pleased to inform you that your manuscript has been judged scientifically suitable for publication and will be formally accepted for publication once it meets all outstanding technical requirements.

Kind regards,

Holly Seale

Academic Editor

PLOS ONE

Additional Editor Comments (optional):

Reviewers' comments:

Reviewer's Responses to Questions

**Comments to the Author**

1. If the authors have adequately addressed your comments raised in a previous round of review and you feel that this manuscript is now acceptable for publication, you may indicate that here to bypass the “Comments to the Author” section, enter your conflict of interest statement in the “Confidential to Editor” section, and submit your "Accept" recommendation.

Reviewer #1: (No Response)

Reviewer #2: All comments have been addressed

2. Is the manuscript technically sound, and do the data support the conclusions?

Reviewer #1: Yes

Reviewer #2: Yes

3. Has the statistical analysis been performed appropriately and rigorously? 

Reviewer #1: Yes

Reviewer #2: Yes

4. Have the authors made all data underlying the findings in their manuscript fully available?

Reviewer #1: Yes

Reviewer #2: Yes

5. Is the manuscript presented in an intelligible fashion and written in standard English?

Reviewer #1: Yes

Reviewer #2: Yes

6. Review Comments to the Author

Reviewer #1: The author's have revised and corrected the manuscript. The manuscript has improved and looks good to get published.

Reviewer #2: Ref: PONE-D-20-11122R1

The critical role of infection prevention overlooked in Ethiopia, only one-half of health-care workers had safe practice: A Systematic Review and Meta-Analysis

Dear Editor

Thank for you for the opportunity to review this manuscript. The study reviews the prevalence of safe prevention practices and the associated factors among healthcare workers in Ethiopia.

My recommendation

My comments have adequately addressed by the authors and hence I recommend this work for next steps!

Thank you

7. PLOS authors have the option to publish the peer review history of their article (what does this mean?). If published, this will include your full peer review and any attached files.

Reviewer #1: **Yes: **Mohammed Owais Qureshi

Reviewer #2: **Yes: **Erick Kinyenje

---

## [Editor Report · Acceptance letter]

6 Jan 2021

PONE-D-20-11122R1 

The critical role of infection prevention overlooked in Ethiopia, only one-half of health-care workers had safe practice: A Systematic Review and Meta-Analysis 

Dear Dr. Sahiledengle:

I'm pleased to inform you that your manuscript has been deemed suitable for publication in PLOS ONE. Congratulations! Your manuscript is now with our production department. 

Kind regards, 

on behalf of

Dr. Holly Seale 

Academic Editor

PLOS ONE